# Reduced Virus Load in Lungs of Pigs Challenged with Porcine Reproductive and Respiratory Syndrome Virus after Vaccination with Virus Replicon Particles Encoding Conserved PRRSV Cytotoxic T-Cell Epitopes

**DOI:** 10.3390/vaccines9030208

**Published:** 2021-03-02

**Authors:** Simon Welner, Nicolas Ruggli, Matthias Liniger, Artur Summerfield, Lars Erik Larsen, Gregers Jungersen

**Affiliations:** 1Section for Veterinary Clinical Microbiology, Department of Veterinary and Animal Sciences, University of Copenhagen, Dyrlægevej 88, 1870 Frederiksberg C, Denmark; lael@sund.ku.dk; 2Institute of Virology and Immunology IVI, Sensemattstrasse 293, 3147 Mittelhäusern, Switzerland; nicolas.ruggli@ivi.admin.ch (N.R.); matthias.liniger@ivi.admin.ch (M.L.); artur.summerfield@ivi.admin.ch (A.S.); 3Department of Infectious Diseases and Pathobiology (DIP), Vetsuisse Faculty, University of Bern, Länggassstrasse 120, 3012 Bern, Switzerland; 4Center for Vaccine Research, Statens Serum Institut, Artillerivej 5, 2300 Copenhagen S, Denmark; grju@ssi.dk

**Keywords:** porcine reproductive and respiratory syndrome virus (PRRSV), virus replicon particles (VRP), classical swine fever virus (CSFV), viral vector, vaccine, polyepitope antigen, cytotoxic T cells, cell-mediated immunity

## Abstract

Porcine reproductive and respiratory syndrome virus (PRRSV) causes severe respiratory distress and reproductive failure in swine. Modified live virus (MLV) vaccines provide the highest degree of protection and are most often the preferred choice. While somewhat protective, the use of MLVs is accompanied by multiple safety issues, why safer alternatives are urgently needed. Here, we describe the generation of virus replicon particles (VRPs) based on a classical swine fever virus genome incapable of producing infectious progeny and designed to express conserved PRRSV-2 cytotoxic T-cell epitopes. Eighteen pigs matched with the epitopes by their swine leucocyte antigen-profiles were vaccinated (N = 11, test group) or sham-vaccinated (N = 7, control group) with the VRPs and subsequently challenged with PRRSV-2. The responses to vaccination and challenge were monitored using serological, immunological, and virological analyses. Challenge virus load in serum did not differ significantly between the groups, whereas the virus load in the caudal part of the lung was significantly lower in the test group compared to the control group. The number of peptide-induced interferon-γ secreting cells after challenge was higher and more frequent in the test group than in the control group. Together, our results provide indications of a shapeable PRRSV-specific cell-mediated immune response that may inspire future development of effective PRRSV vaccines.

## 1. Introduction

Porcine reproductive and respiratory syndrome virus (PRRSV) is a small-enveloped virus with a single-stranded positive-sense RNA genome of 15 kilobases. PRRSV infections cause reproductive failure in late gestation sows [1] and respiratory distress, particularly in young pigs [2]. The level of virulence varies among strains, spanning from very mild symptoms to the detrimental hemorrhagic ‘Porcine High Fever Disease’ caused by highly virulent strains from South-East Asia [3] and the USA [4].

The virus belongs to the *Arteriviridae* family of the order *Nidovirales* and was originally divided into genotypes 1 and 2, representing the European and North American/Asian genotypes, respectively. Recent revision of the *Arteriviridae* taxonomy has reclassified the two genotypes into two distinct species: the Betaarterivirus suid 1 (PRRSV-1) and Betaarterivirus suid 2 (PRRSV-2), respectively [5]. PRRSV-1 is further divided in three subtypes and PRRSV-2 consists of nine lineages [6]. The two species are enzootic in most swine producing countries and cause tremendous production losses worldwide [7,8,9].

Vaccination is the most common method to control the virus. The strongest protective response is obtained using species-specific modified live virus (MLV) vaccines. The use of MLV vaccines, however, has a number of drawbacks: (1) it is well documented that the MLV vaccines may spread to naïve animals, which may end up with enhanced transmission, reversion to virulence, recombination, and disease [10,11,12,13,14]; (2) vaccination of pregnant sows with MLV vaccines in the last trimester may result in reproductive failures, or birth of stillborn and/or persistently-infected piglets [15]; (3) MLV vaccines may persist in a herd for months, or even years, making virus eradication difficult without production stop; 4) MLV vaccines have a limited efficacy against heterologous field strains. Restrictive measures to contain these safety issues have been established. As such, according to the specific product descriptions, MLV vaccines registered in Europe are not approved for use in PRRSV-negative herds and in breeding age boars (ema.europa.eu). Yet, there is an urgent need for alternative PRRSV vaccines to ensure a safe and effective protection against PRRSV.

Multiple vaccine strategies have been tested including killed virus, viral vectors, vaccines based on recombinant protein and DNA with various antigens, delivery systems, and adjuvants. The performance of these vaccines in terms of effect on viral clearance and relief of symptoms are diverse (reviewed in [16,17]). Although they all succeed to induce some degree of an immune response—characterized by virus-specific antibodies and T-cell responses—none of them were capable of providing a sustained protective response against a heterologous challenge. 

Both T-cell responses and especially humoral immunity in response to PRRSV infection have been investigated extensively (reviewed in [18]). The results of these studies are often contradictive and the conclusions regarding the importance of T-cell responses in the protective immune response against PRRSV are vague. It does appear, however, that both neutralizing antibodies and interferon (IFN)-γ play an important role: in one study, passive transfer of virus-specific antibodies provided protection against reproductive failure and sterilizing immunity against a homologous strain, thus, completely bypassing cell-mediated immunity [19]. Another study argued that a T-cell response was solely responsible for the protective immunity of a PRRSV-1 challenge upon vaccination with an MLV vaccine, since a virus-specific IFN-γ response was observed, while no neutralizing antibodies were present [20].

Virus replicon particles (VRPs) represent an RNA vaccine platform that—similar to viral vectors—can induce both humoral and cell-mediated immune responses through sustained RNA replication and expression of vaccine candidate antigens [21,22]. In comparison to virus vectors, VRPs are safer and easier to control as they cannot package their genome into infectious progeny virions unless the missing structural proteins are provided by trans-complementation [23]. Consequently, once a cell has been infected with VRP, the replicon multiplies inside the cell and the genes encoded by the VRP are translated to protein. This activates the endogenous pathway for peptide presentation on major histocompatibility complex (MHC) class I allowing for the generation of a CD8^+^ cytotoxic T lymphocyte (CTL) response without any risk of infectious virus particle formation. VRPs can be regarded as self-adjuvanting [24] since they trigger the innate immune pathways similarly to infectious virus.

Several replicon-based vaccines have been tested in both human and animal trials and have been licensed as commercial vaccines (replicon vaccines reviewed in [25,26]). A recent experiment in the context of PRRSV described the vaccination of pigs with a recombinant vesicular stomatitis virus VRP expressing the PRRSV structural proteins, GP2-5, M, and the nucleocapsid protein N [27]. Although no reduction in viremia was observed following challenge with PRRSV, antibodies against the N protein were detected prior to challenge, and an antibody response against GP3/GP4/GP5 was observed after PRRSV challenge in the VRP-vaccinated animals two weeks earlier compared to the pigs that had received the empty control VRP.

The majority of the studies describe replicons that express whole proteins or larger antigenic fragments, but recent advances in custom DNA synthesis and next-generation sequencing technologies have accelerated replicon development and facilitated the integration of specifically designed gene cassettes. In the present study, we describe the development of a VRP-based vaccine using non-cytopathogenic classical swine fever virus (CSFV) replicons targeting the induction of a sustained and cross-reactive T-cell response against PRRSV-2. To this end, we used classical swine fever (CSF)-VRPs to express nine different polyepitopes resulting from different combinations of a total of 33 conserved PRRSV-2 T-cell epitopes verified previously as binders to relevant MHC class I swine leukocyte antigens (SLA), SLA-1*04:01, SLA-1*07:02, and SLA-2*04:01 [28]. CSF-VRP was chosen as the preferred platform because of the natural tropism of CSFV for antigen presenting cells [29,30], because of the versatility of the platform [31], and because it was used successfully to induce T-cell responses against influenza virus NP [32] and to prime immune responses against PRRSV [27]. We characterized the VRPs in cell culture and in a subsequent vaccination-challenge experiment of young pigs with MHC class I profiles matching the selected epitopes. Our data showed that the VRP-induced T-cell response alone did not protect against infection and disease but resulted in partial reduction of virus load in the lung.

## 2. Materials and Methods

### 2.1. Polyepitope Design and Plasmid Construction

Polyepitopes were designed by a Python-based algorithm encoded to iteratively recombine the individual PRRSV epitopes (previously verified to form peptide-MHC complexes with recombinant SLA I and β2m [28]) in different successions interspersed with random spacer amino acids, meanwhile optimizing for the lowest number of neoepitopes in the regions spanning two neighboring PRRSV epitopes. In this context, neoepitopes were defined as amino acid stretches of 8–11 residues that were predicted to bind to either of 19 SLA class I alleles with a rank ≤4 using the prediction server, NetMHCpan version 2.8. The optimized polyepitopes were reverse translated to cDNA sequences, flanked by a 5′-terminal *Kas*I and a 3′-terminal *Mlu*I restriction site for insertion into the replicon plasmids (see below), and purchased from GenScript (Piscataway) as synthetic gene cassettes codon-optimized for porcine tRNA.

The replicon constructs of the present study were based on the plasmid pA187-N^pro^-IRES-C-delE^rns^ encoding a bicistronic CSFV replicon for transgene expression [31]. This plasmid was derived originally from the full-length cDNA clone pA187-1 [33] by deleting the E^rns^ gene and introducing an internal ribosome entry site (IRES) between the N^pro^ and C genes. For the purpose of the present study, a synthetic gene cassette codon-optimized for porcine tRNA and encoding the C-terminal part of N^pro^ with a C_138_A substitution to abolish inhibition of type I interferon induction by N^pro^, a porcine ubiquitin monomer (Ub, GenBank accession: NP_001098779) mutated to prevent C-terminal cleavage (see below), a hemagglutinin tag (HA), a *Kas*I restriction site, the SIINFEKL epitope (epi), a *Mlu*I restriction site, the FLAG tag, and a stop codon was obtained from GenScript (Piscataway) and used to replace the *Cla*I-to-*Not*I fragment of pA187-N^pro^-IRES-C-delE^rns^. As mentioned, the codon for the C-terminal glycine (G_76_) of the Ub gene was mutated to express a valine (G_76_V) in order to prevent cell-mediated cleavage of Ub from the downstream HA-tagged polyepitope. The resulting plasmid was termed pA187-N^pro^-epi-IRES-C-delE^rns^ and served as a backbone for different PRRSV-2 polyepitopes by replacement of the SIINFEKL epitope with the polyepitope sequence of interest using the restriction sites *Kas*I and *Mlu*I (Appendix A). All final constructs were verified by nucleotide sequencing before they were used to rescue VRPs.

### 2.2. VRP Rescue

The VRPs were rescued from plasmids as described elsewhere [31]. Briefly, plasmids were linearized with the restriction endonuclease *Srf*l and RNA run-off transcription was performed using the MEGAscript T7 kit (Ambion, Austin, TX, USA). One microgram RNA was then used to electroporate 8 × 10^6^ SK-6(E^rns^) cells maintained in Eagle’s minimum essential medium (EMEM) supplemented with 7% horse serum (Håtunalab, Bro, Sweden) and 0.25 mg/mL G418 (Calbiochem, Merck KGaA, Darmstadt, Germany). After three days of incubation at 37 °C, the VRPs were harvested by two freeze-thaw cycles and the lysates were clarified by centrifugation (P0 stocks). The VRPs were further propagated in SK-6(E^rns^) cells by infection at a multiplicity of infection (MOI) of 0.1 followed by incubation at 37 °C for 72 h to generate P1 and P2 stocks.

### 2.3. Titration of VRPs and PRRSV

The VRPs were titrated in SK-6(E^rns^) cells by end-point dilution and immunoperoxidase staining using the anti-E2 monoclonal antibody (mAb) HC/TC26 [34] kindly provided by I. Greiser-Wilke (Hannover Veterinary School, Hannover, Germany) and a horseradish peroxidase (HRP)-conjugated rabbit anti-mouse IgG (Dako). Alternatively, the VRPs were titrated in PK-15 cells using the mAb WH211 (APHA, RAE0242) and an HRP-conjugated polyclonal goat anti-mouse serum (Jackson ImmunoResearch Laboratories, West Grove, PA, USA). Titration of the PRRSV-2 strain used for challenge was performed in MARC-145 cells using the monoclonal antibody, SDOW17-A (RTI LLC) and HRP-conjugated rabbit anti-mouse serum (Dako). The SK-6(E^rns^) cells were maintained as described above, and the PK-15 and MARC-145 cells were cultured in Dulbecco’s modified Eagle medium supplemented with 5% fetal bovine serum (FBS).

### 2.4. Flow Cytometry

Flow cytometry (FCM) was applied to VRP-infected cells for the detection of the FLAG-tagged epitope expression. Briefly, 10^5^ SK-6 cells were infected with the VRPs 0 to 9 from the first passage (P1) stock or mock infected. Mock consisted of SK-6(E^rns^) lysate obtained in parallel to the VRP stocks. In addition, VRPs rescued from the original backbone replicon vector, pA187-N^pro^-IRES-C-delE^rns^, were included as a negative sample control as this replicon does not encode a FLAG tag. All infections were performed at a MOI of 5 in the presence of 100 nM of the proteasome inhibitor epoxomicin (Sigma) or of an equivalent amount of dimethyl sulfoxide (DMSO) as solvent control added 28 h post-infection, in order to counter the expected proteasomal degradation of the ubiquitinylated polyepitopes. After another 18 h, the cells were detached by trypsin treatment, fixed with 4% paraformaldehyde in phosphate buffered saline (PBS), and permeabilized with 0.1% saponin in PBS. Infection was confirmed by the detection of the CSFV E2 protein with the mAb HC/TC26 and AlexaFluor647-conjugated goat anti-mouse IgG2b (ThermoFisher, Waltham, MA, USA). Polyepitope expression was confirmed by FLAG tag detection with the F3165 mAb (Sigma, Kawasaki, Japan), and the phycoerythrin-conjugated goat anti-mouse IgG1 (BioConcept, Allschwil, Switzerland). All antibodies were diluted in PBS + 0.3% saponin. The cells were washed with Cell Wash (BD Biosciences) after each treatment and subjected to FCM (FACSCanto II, BD Biosciences).

### 2.5. SLA Genotyping

Sequence-specific SLA genotyping was performed by polymerase chain reaction (PCR) on genomic DNA extracted from ethylenediaminetetraacetic acid (EDTA)-stabilized whole blood from candidate experimental animals using primers specific for the supertypes SLA-1*04 (forward: 5′-GCCTGACCGCGGGGACTCT-3′, reverse: 5′-CTCATCG-GCCGCCTCCCACTT-3′), SLA-1*07 (forward: 5′-GCCGGGTCTCACACATCCAGAT-3′, reverse: 5′-GGCCCTGCAGGTAGCTCCTCAAT-3′) and SLA-2*04 (forward: 5′-CCGAGGGAACCTGCGCACAGC-3′, reverse: 5′-CCCACGTCGCAGCCGTACATGA-3′). Amplicons were sequenced by commercial Sanger sequencing (LGC Genomics) and identification of the alleles, SLA-1*04:01, SLA-1*07:02, and SLA-2*04:01, was performed by single nucleotide polymorphism (SNP) analysis using CLC Main workbench 7.0.

### 2.6. Experimental Animals

Peptide-MHC I complex formation of the predicted epitopes included in the VRPs were demonstrated experimentally with SLA-1*04:01, SLA-1*07:02, and SLA-2*04:01. Therefore, only animals expressing these alleles were included in the polyepitope vaccination and PRRSV-2 challenge trial. Thirteen pregnant sows from a Danish herd certified and verified by serology to be free of PRRSV (results not shown) were genotyped. Four of these were found to carry at least one of the three SLA alleles of interest. All 45 piglets of the four sows (offspring from Danish Landrace-Yorkshire sows crossed with Duroc boars) were genotyped, from which 18 SLA-matched piglets (8 females and 10 males) were selected as experimental animals and purchased. The 4-week-old pigs were all housed in the same pen in the biosafety level 3 agricultural (BSL-3Ag) animal isolation facility at the National Veterinary Institute, Lindholm. Here, they were divided randomly into a test group (N = 11) and a control group (N = 7), with blocking for an even distribution of SLA-profile, litter of origin, and initial bodyweight (Table 1). Ear tag numbers were used to identify the animals in order to ensure blinding (see below). The names were assigned after the end of the experiment for easier distinction of test and control pigs. Seven weeks after arrival, the pigs were separated in two pens with an even distribution of test pigs and control pigs in each pen. Throughout the whole experiment the pigs had free access to water and were fed on a daily basis with zinc-supplemented fodder purchased together with the pigs (first two weeks) or Porkido 10,5 Ideal AU (DLG) (rest of period).

### 2.7. Experimental Setup of the In Vivo Study

The pigs in the test group were vaccinated with a titer-adjusted mix of the PRRSV polyepitope containing VRPs (VRPs 1 to 9), while the pigs in the control group received the VRP encoding the SIINFEKL control peptide (VRP 0, see Appendix A). Vaccinations were administered as intradermal injections (27G needle) of 0.5 mL 10^7^ median tissue culture infectious dose (TCID_50_)/mL VRP from the P2 stock applied as five spots of 0.1 mL each in the dermis of the right-side lateral neck region. The first vaccinations were given one week after arrival at days post vaccination (dpv) 0. This was followed by two booster vaccinations at dpv 28, and 51. Challenge virus was administered intranasally with 2 × 10^6^ TCID_50_/animal of the Danish PRRSV-2 field isolate, DK-1997-19407B (cluster 5.2, GenBank accession KC862576), by spraying 2 mL virus solution into each nostril using a syringe. Challenge was given at dpv 64 (Figure 1). The assignment of the pigs to vaccine or control group remained unknown to the caretakers in charge of clinical evaluation throughout the whole experiment.

### 2.8. Clinical Observations

The pigs were monitored twice daily and a clinical score was assigned based on general health condition (normal, mild lethargic, lethargic, or apathetic), respiration (normal, increased respiration, respiratory distress, severe respiratory distress), and appetite (normal, slow eating, not eating). Body weight was measured at dpv −1, days post challenge (dpc) −1, and at necropsy. Rectal temperatures were measured at dpv −1, 1, 2, 3, 28 (2nd vaccination), 29, 30, 31, 51 (3rd vaccination), 52, 53, 54, and 63 (one day before challenge, dpc −1), and at dpc 1, 2, 3, 5, 7, 8, 9, 13, and 20.

### 2.9. Blood and Nasal Swab Sampling

Blood samples were collected from the jugular vein of all pigs at dpv 0, 14, 21, 41, 51; and dpc −1, 1, 2, 5, 7, 9, 13, and 20 for the preparation of serum and/or peripheral blood mononuclear cells (PBMCs). The samples at dpv 0 were collected prior to vaccination. Serum was recovered from non-stabilized tubes after coagulation overnight at 4 °C by centrifugation at 1000× *g* for 10 min at 4 °C and stored at −80 °C for subsequent analysis. PBMCs were isolated from heparin-stabilized tubes by density centrifugation on Lymphoprep (Stemcell) in 50 mL SepMate tubes (Stemcell) at 1200× *g* for 10 min at room temperature (RT). Contaminating erythrocytes were lysed with lysis buffer (77 mM NH_4_Cl, 5 mM KHCO_3_, 63 μM EDTA in water) for 3 min at RT and washed with PBS + 2% FBS. PBMCs were used the same day for immunological examination by IFN-γ enzyme-linked immunospot (ELISPOT) assay after being counted by microscopy. Nasal swabs were collected from all pigs at dpc −1, 1, 2, 5, 7, 9, 13, and 20 and placed in cryotubes containing 1 mL PBS and stored at −80 °C for subsequent analysis.

### 2.10. Tissue Sampling

Euthanasia was performed at dpc 26 and 27 by captive bolt stunning followed by exsanguination by cutting the *vena* and *arteria axillaris*. Immediately thereafter, a visual inspection of the lungs was performed, and three samples of approximately 1 cm^3^ of lung tissue (left cranial, medial, and caudal lobes) were collected from each pig and kept at −80 °C for subsequent analysis. Additionally, the draining lymph node of the vaccination site, *Ln cervicalis superficialis dorsalis*, was excised from all pigs and cells were extracted manually, separated from debris through a 100 μm cell strainer and washed with PBS + 2% FBS. The cells were used the same day for immunological examination by IFN-γ ELISPOT after being counted under a microscope.

### 2.11. Serology

The detection of serum antibodies against the CSFV E2 glycoprotein was performed with serum from dpv −1 and 51 using a classical swine fever E2 competition enzyme-linked immunosorbent assay (ELISA) kit (CSFE2C-5P, ID-vet) following the manufacturer’s instructions. The plates were analyzed at 450 nm using a ELx808™ absorbance microplate reader (BioTek). The detection of serum antibodies against the PRRSV nucleocapsid protein N was performed with serum from dpc −1, 7, 9, 13, and 20 using an IDEXX PRRS X3 Ab Test (99-40959, IDEXX) following the manufacturer’s instructions with the sole modification that the tetramethylbenzidine (TMB) color reaction was stopped with equivalent amounts of 1M H_2_SO_4_ (in house), after which the plate was read at 450–630 nm using the ELx808™ absorbance microplate reader. Positive and negative controls were measured in duplicates, while samples were performed in single measurements.

### 2.12. IFN-γ ELISPOT

MultiScreen IP filter 96-well plates (Millipore, MSIPS4510) were treated with 35% EtOH for <60 s and coated with 250 ng/well mouse anti-porcine IFN-γ monoclonal antibody (clone P2F6, ThermoFisher) in PBS at 4 °C overnight. Plates were washed three times in PBS and blocked with AIM-V albuMAX (31035025, ThermoFisher, Waltham, MA, USA) at 37 °C, 5% CO_2_ for at least 1 h after which freshly isolated cells were seeded in presence of stimuli as described below. Peptides used for stimulation (GenScript, Piscataway, NJ, USA) were dissolved and stored as described in Appendix A. Following two days of incubation at 37 °C, 5% CO_2_, the plates were emptied and the cells were lysed by two times washing with ultrapure water, then three times with washing buffer (PBS + 0.01% Tween 20). Plates were incubated with 100 ng/well biotinylated mouse anti-porcine IFN-γ mAb (clone P2C11, BD Biosciences) in reaction buffer (PBS + 0.01% Tween 20 + 0.1% bovine serum albumin) on a shaker at RT for 1 h. The plates were washed four times and incubated with 50 mU/well streptavidin-alkaline phosphatase-conjugate (11089161001, Sigma-Aldrich, St. Louis, MO, USA) in reaction buffer on a shaker at RT for 1 h. Plates were washed three times with washing buffer followed by two times with PBS. Spots were developed using 100 μL/well BCIP/NBT tablets (B5655, Sigma-Aldrich, St. Louis, MO, USA) dissolved in 10 mL/tablet ultrapure water in the dark at RT for 5 min, and the development was stopped under running tap water while the underdrain was removed. Still wet, the plates were completely submerged in decontamination solution (1% VirkonS) for 30 min prior to export from the BSL-3Ag facility in compliance with the biosafety regulations. The plates were washed under running tap water and left to dry in the dark. Ultimately, the spots were counted on an AID iSpot Reader Spectrum (Autoimmun diagnostika GmbH).

#### 2.12.1. Pre-Challenge ELISPOT

ELISPOT assays using PBMCs isolated at dpv 0, 14, 27, 41, and 63 were designed to screen for reactive peptides included in the VRPs pre-challenge. Twelve peptide-pools were used for restimulation of the PBMCs representing a two-dimensional matrix with six pools in each dimension containing five to six peptides each. Together, each of the 33 PRRSV peptides included in the VRPs was represented by exactly one pool in each dimension. Stimulations with the VRP mixture used for vaccination and the PRRSV strain used for challenge were also included together with their respective mocks. Peptide stimulations were done with partial concentrations of 2 μM/peptide, while virus and VRP stimulations were done at a MOI of 0.1. Unstimulated wells were included as baseline, and wells stimulated with 1 μg/mL staphylococcal enterotoxin B (SEB) (S4881, Sigma-Aldrich) were included as positive controls. All stimulations were seeded in duplicates with 300,000 cells/well.

#### 2.12.2. Post-Challenge ELISPOT

ELISPOT assays using PBMCs isolated at dpc 7 and 20 were designed to identify individual reactive peptides among 14 peptides chosen from the 33 vaccine peptides. Stimulations were done with individual peptides at concentrations of 5 μM. Unstimulated wells were included as baseline, and wells stimulated with 1 μg/mL SEB were included as positive controls. All stimulations were seeded in quadruplicates with 500,000 cells/well. Consequently, restimulation with virus and VRP were excluded from the setup due to limitations in test capacity and number of PBMCs available. This setup was also used for the ELISPOT assays using cells extracted from the lymph nodes, although with 300,000 cells/well only.

### 2.13. Quantification of Viral RNA

Viral RNA was purified from the challenge inoculum, serum samples, nasal swabs, and lung tissue homogenate. The samples were clarified by centrifugation and RNA extracted using the MagNA Pure LC Total Nucleic Acid Isolation Kit (03 038 505 001, Roche, Basel, Switzerland) on a MagNA Pure LS Instrument (Roche, Basel, Switzerland). Quantitative reverse transcription PCR (qRT-PCR) was performed using the Qiagen OneStep RT-PCR Kit (210210, Qiagen, Hilden, Germany) on an MX3005P QPCR System (Agilent) with the following primers: forward: 5′-ATRATGRGCTGGCATTC-3′, reverse: 5′-ACACGGTCGCCC-TAATTG-3′. The probe was modified from the TEX-containing version to contain HEX instead: 5′-(HEX)-TGTGGTGAATGGCACTGATTGACA-(BHQ2)-3′ [35,36]. Cq values were converted to equivalents of TCID_50_ (TCID_50_eq) using a standard curve based on a purified 10-fold dilution series of the challenge isolate. qRT-PCR was performed in duplicates for all samples. Prior to purification, lung tissue homogenate was prepared from cutouts of approximately 0.2–0.4 g of tissue homogenized in 1 mL EMEM using lysing matrix D (MP bio) in a FastPrep FP120 cell homogenizer (Thermo Savant) for 60 s at speed 5.

### 2.14. Statistics

Positive signals upon restimulation in the ELISPOT data were identified by two criteria: the first using the online (http://www.scharp.org/zoe/runDFR/, accessed September 2016) non-parametric distribution free resampling (DFR) tool as described by Moodie et al. [37], and the second by defining a positive signal as a response with more than twice the number of signal spots compared to the number of background spots with a minimum of eight signal spots (ratio-2 method) [38]. This method furthermore allowed for a quantitative analysis of the response magnitudes. The results of the two methods were used to calculate relative percentage values of the responding peptides in order to compare responders between groups.

*p*-values for the differences in lung tissue virus load between groups were calculated using Mann–Whitney. A paired, two-tailed T test was used to test for significant peaks in rectal temperature within the groups pre-challenge. An unpaired, two-tailed T test was used to test for significant difference in rectal temperature between the groups post-challenge.

## 3. Results

### 3.1. Expression and Proteasomal Degradation of the VRP-Encoded PRRSV Polyepitopes

#### 3.1.1. The PRRSV-2 Polyepitope Ensemble

The 33 individual epitopes used in this study were conserved among PRRSV-2 strains and were selected from a large specific epitope pool based on in silico-predicted and in vitro-verified binding-affinity and -stability to three relevant SLAs, as determined in a previous study (Table 2) [28]. They were assembled into a total of nine polyepitopes that were designed in pseudo-triplicates for each of the three SLAs. Pseudo-triplicate means that for each SLA, three polyepitopes were designed with the same individual SLA-specific epitopes, but in different successions to obtain a more robust expression of all epitopes against potential translation and/or degradation artifacts related to primary structures. As such, the pseudo-triplicates for the construction of VRP 1 to 3 were based on epitopes specific for SLA-1*04:01, pseudo-triplicates for VRP 4 to 6 were based on epitopes specific for SLA-1*07:02, and pseudo-triplicates for VRP 7 to 9 were based on epitopes specific for SLA-2*04:01 (Table 3).

#### 3.1.2. VRPs Designed to Feed Ubiquitinated PRRSV Epitopes into the MHC-I Presentation Pathway

For the generation of VRPs expressing the polyepitopes described above, the backbone plasmid pA187-N^pro^-epi-IRES-C-delE^rns^ (Figure 2) was derived from pA187-N^pro^-IRES-C-delE^rns^ encoding a bicistronic CSFV replicon used previously to express bioactive luciferase and granulocyte macrophage colony-stimulating factor in SK-6 cells [31]. This backbone vector was modified to contain a synthetic DNA cassette encoding the SIINFEKL epitope as a control (VRP 0) that was replaced with the individual polyepitope cassettes of interest described above to generate VRP 1 to 9. As such, the polyepitopes were expressed as ubiquitin-linked and FLAG-tagged proteins. The C-terminal glycine of Ub was mutated to a valine (G_76_V) in order to prevent cell-mediated cleavage of Ub from the downstream polyepitope by ubiquitin C-terminal hydrolases. This has previously been shown to ensure poly-ubiquitination and proteasomal degradation of the Ub-polyepitope chimera, thereby favoring MHC-I-mediated peptide presentation [39]. The C_138_A substitution in N^pro^ prevents N^pro^-mediated inhibition of type I IFN induction, which is expected to confer adjuvant activity to the replicons in vivo through innate immune activation [31,40]. The annotated details of the nucleotide and amino acid sequences for the backbone plasmid and the nine polyepitopes are available in Appendix A.

#### 3.1.3. Verification of Polyepitope Expression and Proteasomal Degradation in Cell Culture

The backbone plasmid was used for the rescue of VRP 0 expressing the SIINFEKL control epitope, and the constructs encoding the nine individual polyepitopes resulted in VRP 1 to VRP 9, respectively (see Table 3). FCM analysis of cells transduced with VRPs at a MOI of 5 confirmed VRP expression by CSFV E2 detection, and the tagged polyepitope expression and proteasomal degradation were analyzed by FLAG tag detection under epoxomicin or DMSO solvent control treatment. The results are summarized in Figure 3A with a representative example of the FCM gating strategy for VRP 6 shown in Figure 3B.

The threshold for infectivity (E2 detection, horizontal line) was defined based on the epoxomicin-treated mock-infected sample, and the threshold for FLAG tag detection (vertical line) was defined based on the epoxomicin-treated sample infected with the FLAG-negative VRP rescued from the original plasmid, pA187-N^pro^-IRES-C-delE^rns^ (Appendix A). All VRPs were highly infectious and resulted in approximately 80% of E2-positive cells. Polyepitope expression was detected for all constructs in 1% (VRP 9) to 22% (VRP 5) of the cells analyzed, but only when proteasomal degradation was inhibited with epoxomicin (Figure 3A and Appendix A). The FLAG-tagged polyepitope expression with VRPs 7 to 9 was remarkably weaker (1–2% of all cells) compared with VRPs 0 to 6 (5–22% of all cells). This was an interesting observation, as the VRPs 7 to 9 represent SLA-2*04:01-specific polyepitopes as opposed to the other epitopes that are specific for the other two SLAs. Collectively, these results show that all infections were successful, and indicate that the polyepitope expression, as expected from the Ub linkage, undergoes the proteasomal degradation necessary for peptide-MHC class I complex formation and peptide presentation. If one assumes that the epoxomicin treatment does not provide complete proteasomal inhibition, one may postulate that the proportion of transduced cells expressing the polyepitope may be higher than what we measured in our assay.

### 3.2. The Vaccination-Challenge Trial

#### 3.2.1. Clinical Monitoring during the Vaccination and Challenge Phase

No virus-related clinical signs were seen in any of the pigs during the experimental period and all pigs had a normal weight gain (data not shown). However, moderate but statistically significant variations in rectal body temperature (up to 40.6 °C) did reflect the second and third vaccination events. Following challenge, no difference in rectal temperature between the two groups was observed (Figure 4A). This is in accordance with previous studies using this field strain [41].

#### 3.2.2. Seroconversion Demonstrated In Vivo Replication of Both the Replicon Vaccine and the Challenge Virus

Two vaccinations at 28 days interval were sufficient to induce a strong seroconversion against the vaccine vector as measured with an anti-E2 competition ELISA on day 51 post-vaccination, immediately before the third VRP injection (Figure 4B; optical density (OD) span 9–29%, positive < 50), while all pigs were clearly seronegative for E2 before the first vaccination (Figure 4B; OD span: 76–148%, negative > 60). Due to the observation made by Suter et al. [31] that only pigs immunized with live CSF-VRP seroconverted towards E2, whereas pigs immunized twice with UV-inactivated CSF-VRP did not, our results indicated that the VRP-encoded genes were efficiently transcribed and translated into protein in the vaccinated animals. However, the vaccination had no effect on the kinetics of the PRRSV-specific antibody response against the nucleocapsid protein N after challenge as shown with the semiquantitative IDEXX PRRS X3 Ab Test kit on days −1, 7, 9, 13, and 20 post-challenge. All animals mounted an antibody response against PRRSV from day 7 post-challenge, with no difference between the groups (Figure 4C).

### 3.3. The Polyepitope Vaccination Induced Peptide-Specific T-Cell Responses

With PBMC collected during the vaccination period before challenge infection the IFN-γ ELISPOT revealed non-specific spots in all wells including the non-stimulated controls, thereby making it impossible to identify clearly peptide-specific signals (data not shown). An improved IFN-γ ELISPOT setup was performed with PBMC collected on days 7 and 20 after challenge and in part with lymph node cells obtained at necropsy. A higher resolution was obtained from this setup by increasing the number of replicates and the number of cells per well. In addition, only individual peptides and at higher concentrations were used for restimulation instead of peptide pools. As a tradeoff, only 14 peptides could be included in the assay, and restimulations with virus and VRP were excluded due to limitations in the number of PBMCs available. The 14 selected peptides (ID 2, 7, 12, 13, 19, 21, 23, 24, 25, 27, 28, 36, 39, and 44) were chosen based on their in vitro binding capacities (see Table 2). Figure 5 shows the restimulation signals (ELISPOT counts) for each individual animal and peptide, and Figure 6 shows a compilation of these results per group. The results are interpreted using the two statistical methods ratio-2 and DFR. Their results are mostly overlapping, but while DFR is a well-established statistical method it only provides a qualitative output, whereas a quantitative comparison can be obtained with the ratio-2 method as seen in the top panel of Figure 6A. For either method, significant responders are indicated in the middle and bottom panels of Figure 6A, respectively. Regardless of the method, the test group clearly shows more frequent and stronger responses of the PBMC compared to the control group, thereby suggesting an effect of the vaccine in the induction of a T-cell response. The lymph node cells were overall much less responsive than the PBMCs with only three responder pigs out of 18. Nevertheless, these were all from the test group. The general response-dominance of the test group still prevailed after accounting for the unequal number of pigs in the two groups, as presented in Figure 6B.

It is worth noting that pig “Toby” responded to peptide 44 (MSWRY**S**CTRY) at dpc 7, although the peptide had a mismatch compared with the corresponding sequence encoded by the PRRSV challenge strain (MSWRY**A**CTRY). It is unlikely that this peptide-specific response was induced by the challenge virus, since in that case, the A_6_S polymorphism would introduce a hydroxylic group of serine in the T-cell receptor (TCR)-binding middle part of the peptide. This substitution would supposedly be detrimental for the binding of a TCR primed by the flat and aliphatic alanin-containing challenge version. This is a good indication that the response originated from the VRP-encoded peptide rather than from the virus.

The detailed analysis revealed that despite the general response-dominance of the test group, surprisingly no animal responded to the same peptide twice, i.e., at 7 and 20 dpc, according to the DFR method, and only one animal (“Thomas”, peptide 28) did so according to the ratio-2 method. This indicates that peptide-specific responses were not dominated by single clones but covered several peptides with low response levels.

In summary, the pigs of the test group have a generally higher response frequency and magnitude than the pigs of the control group, thereby indicating a vaccine-mediated induction of a T-cell response.

### 3.4. The Polyepitope Vaccination Did Not Prevent Viremia, But Resulted in Reduced Virus Load in the Lung

In order to determine the protective potential of the VRP-mediated polyepitope vaccination, the PRRSV load was analyzed by qRT-PCR in nasal swabs and serum on days 5 and 13 after challenge. The nasal swab samples were combined in one pool per group while the sera were tested individually. Very low levels of virus were detected in the nasal swabs, with no apparent trend between groups (data not shown). In the serum samples collected on day 5 and 13 post-infection, the TCID_50_ equivalents as determined by qRT-PCR were overall higher, but there were no differences between the groups (Figure 7A), showing that the vaccination did not decrease the level of viremia. The virus load was also determined in lung tissue samples collected from the cranial, middle, and caudal parts of the lung. Here, slight differences in TCID_50_ equivalents per gram tissue between the groups were observed in the middle parts (*p* = 0.069), and significant differences were found in caudal (*p* = 0.035) parts of the lungs, which may be attributed to the vaccination (Figure 7B).

In general, PRRSV-specific T-cell response is not detected within the first four weeks after infection [42,43,44,45]. Seen in that light, five test-group pigs stand out from the remainders of the group by having responded clearly to peptides already seven days after infection according to either the ratio-2 method alone (“Tyson”) or both methods (“Toby”, “Trisha”, “Thomas,” and “Tina”). This suggests that the IFN-γ ELISPOT responses at day 7 post-challenge were primed by the vaccination. Revisiting the viral load data in this new context did again not reveal any differences in viremia between the groups (data not shown), but substantiated the differences of viral load in the lungs. This was particularly the case in the middle part of the lung, where a significant difference was observed when these early responders were compared with the rest of the test-group pigs that did not show a T-cell response on dpc 7 (Figure 7C). This correlation between reduced viral load and early T-cell response is indicative of a vaccine-induced protective response.

## 4. Discussion

The present study explored a rational approach for the induction of PRRSV-specific T-cell response through vaccination with CSF-VRPs expressing conserved PRRSV-2 MHC class I epitopes selected from a previous study [28]. The epitopes and experimental animals were matched in terms of MHC class I-restriction and -profile, respectively, and the VRPs were constructed to induce ubiquitination and endosomal processing of peptides for optimal presentation in peptide-MHC class I complexes. While the vaccination had no effect on the prevention of infection and viremia, there was evidence of a virus load reduction in the lungs, suggesting a contributing protective effect of a VRP-mediated polyepitope-induced T-cell response.

The rationale behind choosing the VRP technology as vaccine platform was the fact that it clearly avoids the major drawbacks, such as vaccine virus spread, recombination and reversion to virulence, and the risks of reproductive failures and weak-borne piglets associated with MLV application during the last trimester of gestation. Due to the replicative properties of the VRP, their tropism for antigen presenting cells, and the replicon-mediated innate immune activation, VRPs are capable of inducing a good immune response, meanwhile maintaining absolute safety and high adaptability through simple and flexible genetic engineering [29,31]. Several examples already exist for other viruses where replicon- or viral vector-based vaccines expressing transgenic single epitopes or polyepitopes were capable of inducing a CTL response. In many cases, protective immunity upon challenge has even been successfully established [46,47,48,49]. The CSF-VRP, like the virus they are derived from, have a tropism for dendritic cells (DC) and both conventional DCs (cDC) and plasmacytoid DCs (pDC) are early targets for the virus [29]. During normal CSFV infections, both the N^pro^ and E^rns^ proteins of CSFV prevent type I IFN induction, respectively, by targeting IFN regulatory factory 3 for degradation [40,50] and by interfering with Toll-like receptor 7-mediated pDC activation in yet unknown intracellular compartments, a mechanism that is dependent of the RNase activity of E^rns^ [51]. In the CSF-VRP applied here, these activities were both abolished by a mutation in N^pro^ and by deletion of E^rns^. A previous study in pigs vaccinated intradermally with A187-delE^rns^, a CSF-VRP related to the VRP used here, showed a clear T-cell response as measured by increased IFN-γ production after ex vivo re-stimulation of T-cells with CSFV [52]. In a study attempting to determine the source of CSFV-induced IFN-γ, it was demonstrated that CD3^+^CD4^-^CD8α^high^, consistent with CTLs, were the initial source of CSFV-specific IFN-γ producing cells upon challenge of animals vaccinated with an attenuated CSFV C-strain. In contrast, no T-cell IFN-γ was detectable upon challenge of unvaccinated animals that developed clinical signs of disease [53]. A CSF-VRP based vector vaccine was also shown to induce both a CD4 and CD8 T-cell response against nucleoprotein of influenza virus, confirming the suitability of VRP to induce cell-mediated immunity [22,32].

In the present study it was hypothesized that a PRRSV-specific T-cell response, as indicated by PRRSV-induced IFN-γ secretion, should result from CTLs activated via their TCRs by cognate PRRSV peptide-MHC complexes presented by antigen-presenting cells. As such, the central concept of the study was to use only peptide-MHC combinations that had previously been identified as stable binders [28]. Consequently, only experimental animals expressing at least two of the SLA alleles of interest were included in the animal experiment. Out of the 33 selected epitopes, three sets of three SLA-specific polyepitopes were created in pseudo-triplicates. The purpose of the SLA-matching polyepitopes was to allow ‘individualized’ vaccination by administration of a mixture of polyepitopes corresponding to the SLA profiles of the individual animals. However, this strategy could only be partially fulfilled in the present study. The design of pseudo-triplicates was to ensure a more stable expression of all epitopes against potential translation and/or degradation artifacts related to primary structures. The nine different polyepitopes were inserted into the expression cassette of the CSFV replicon by replacing the SIINFEKL control epitope. The cassette had two important features: firstly, a non-cleavable ubiquitin molecule upstream of the epitope site was expected to feed the polyepitope into the MHC-I presentation pathway via the immunoproteasome [39], and secondly, a FLAG tag downstream of the epitope served to detect polyepitope translation. Both features were verified successfully for all VRPs by FCM on infected SK-6 cells.

Indications of both vaccine-specific T-cell responses and reduced viral load in the lungs were observed in our results. In spite of this, the protective effects were only partial and the T-cell response readouts were highly variable and of low magnitude. This is consistent with previous findings in which a recombinant adenovirus vector expressing PRRSV polyepitopes was successfully shown to induce significant epitope-specific IFN-γ responses in a vaccine-challenge experiment [54]. In this latter study, this was not sufficient to confer full protection, but nevertheless, the vaccinated animals showed a higher challenge virus clearance rate than the sham-vaccinated animals.

In our study, the reasons for only partial protection induced by the vaccine may be related to insufficient VRP-induced priming of the CTLs. The chain of events from vaccination to CTL priming is long and involves several steps subject to potential erroneous processing that may ultimately result in unsuccessful priming. An important first step is the activation of DCs for subsequent migration to, and antigen presentation in the lymph nodes [55]. This step is unlikely to be the cause of unsuccessful priming due to the features of the CSF-VRP platform described above and to the evidence of strong antibody responses to the CFSV E2 protein. Thus, assuming that infection and activation of antigen presenting cells was not the bottleneck of a more consistent and strong T-cell response, failures in intracellular processes related to polyepitope expression and processing may be responsible. In this context, epitope abundance and insufficient peptide-MHC complex stability are relevant parameters. This is evident by the low expression levels shown with FCM and by the fact that immunoblot analysis of lysates of cells treated in vitro with epoxomicin and infected with VRP failed twice to show visible bands (data not shown). Regardless, the vaccination experiment was continued as intended, since vector DNA sequencing confirmed the presence of the expected cassette and flow cytometric analyses of VRP-infected SK-6 cells indicated correct polyepitope expression. Of note, all analyses indicated low levels of protein expression, which was, however, not considered a hindrance to CTL priming. Following translation, the amount of individual epitopes would have decreased further upon proteasomal degradation and N-terminal trimming by aminopeptidases, processes that would undoubtedly eliminate a fraction of the epitopes. One study indicated that the sets of peptides produced by either the conventional or the immunoproteasome differ more than expected [56]. This is highly relevant due to the fact that activated DCs mainly contain the latter. Ultimately, the peptide-MHC complex stability of the selected epitopes may for some of the peptides not have been sufficiently high to maintain complex formation long enough for T-cell encounter and recognition to occur. The combination of these aspects could have resulted in a very low rate of peptide-MHC-TCR encounters on the surface of infected cells, which may explain the poor immune priming observed. We measured systemic T-cell responses in PBMCs, but this may not fully reflect the local T-cell responses to the challenge infection in the airways. Analysis of local resident memory T cells obtained by bronchoalveolar lavage (BAL) would most likely have increased numbers of PRSSV-specific T cells as recently shown with swine influenza [57]. Indeed, CTLs have been reported to play a crucial role in combatting PRRSV at the site of infection, in lungs and in BAL [58].

An alternative or additional cause of the low T-cell responses observed in our study could be that the primed CTL response after vaccination was inhibited by the PRRSV challenge infection. PRRSV is notoriously known for its multiple immunoevasive mechanisms (reviewed in [59]) among many others are the downregulation of SLA-I molecules on the surface of infected cells [60], the increased secretion of interleukin-10 (IL-10), and the activation of regulatory T-cells, that could drive primed CTLs into quiescence. The low quality of pre-challenge ELISPOT precludes a valid analysis of T-cell responses to vaccination only and PRRSV-induced immunosuppression could explain at least part of the low IFN-γ responses and the weak effect on virus load after challenge.

It is interesting that five test animals showed increased viral clearance in the lungs when compared to the remainders of the test group. As dictated by the central concept that a T-cell response would be the result of stable peptide-MHC bonds, all the selected epitopes were verified as binders to either of the three SLA alleles. As such, the animals included in the experiment did all express at least two of these alleles. The screening method to identify these animals was based on sequence-specific SLA genotyping of only these three SLA alleles and not by complete SLA genotyping. It is, thus, a possibility that the five animals with increased viral clearance had an untyped SLA allele in common that was capable of mounting a strong T-cell response via interactions with one or more of the epitopes included. In general, pigs express a paternal and a maternal allele of each of the three SLA class I loci, meaning that each pig expresses between three and six unique alleles. In our case, the pigs were offspring of mixed races, thereby increasing the likelihood of them expressing six rather than three unique alleles, which could certainly play a role in the big variations of induced T-cell responses.

The large diversity of SLA haplotypes is an evolutionary advantage to avoid escape mutants of the virus, but imposes a challenge with respect to developing a rational vaccine platform based solely on CTL epitopes. One way to approach broad SLA coverage is to develop a library of SLA-specific polyepitopes, of which singlet polyepitopes can be combined into an ‘individualized’ vaccine shot exactly matching the SLA-profile of the target animal. This was the approach pursued in the present study. An alternative approach is to develop a more general and uniform CTL epitope-based vaccine consisting of an ensemble of carefully selected epitopes that in combination will cover the diversity and individual abundance of both viral strains and SLA haplotypes. The PopCover algorithm does exactly that and was also used to select the epitope candidates for this study [28,61]. A major limitation of this approach—and the reason it was not used in the present study—is the enormous amount of labor required to validate the predicted peptide-SLA binding capacities in vitro. Regardless, a full SLA profile of the experimental animals could have revealed patterns correlating with the observations, why full SLA genotyping is encouraged for future studies of this type.

This study describes a rational approach for the induction of a PRRSV-specific CTL response via vaccination with VRPs expressing conserved PRRSV-2 MHC class I T-cell epitopes and deliberately omitting a humoral antibody response to PRRSV structural epitopes to dissect the importance of T-cell based immunity. Although the response magnitude was lower than expected, our results suggest that a T-cell response was established and that some degree of protection was obtained in five out of the 11 test pigs. The challenge strain used in the present study belongs to the clade 5.2, which is the only clade circulating in Europe. Despite that this strain causes severe clinical signs in the field, it is difficult to reproduce severe clinical signs and extended viremia under experimental conditions [41]. It is possible that the differences between the vaccinated and unvaccinated animals would have been clearer if a more pathogenic strain were used for challenge. However, a clear relationship between pathogenicity/virulence and induction of T cell responses has not been established [58]. Ultimately, our attempt to boost the CTL priming by administering booster vaccinations may have been impaired by antibody responses against the structural protein E2 used as positive control for seroconversion in the present setup. In future trials, CSF-VRPs lacking all of the structural protein genes may yield more efficacious booster responses. In any case, in a final setup, CSFV E2 needs to be excluded from any PRRSV vaccine to prevent interference with CSF surveillance, and for a vaccine with better protective efficacy, relevant epitopes for protective antibody responses to PRRSV needs to be included. With this, it can be concluded that CSF-VRP represent a potential platform for epitope vaccination to induce a T-cell response with protective properties, but further optimization of epitope selection and delivery is needed.

## Figures and Tables

**Figure 1 vaccines-09-00208-f001:**
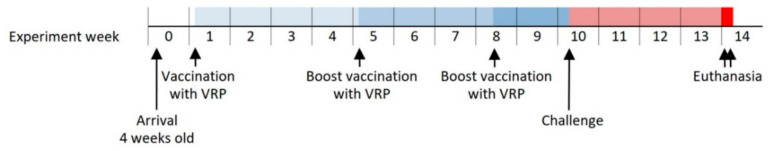
Timeline of the vaccine-challenge experiment indicating the major interventions.

**Figure 2 vaccines-09-00208-f002:**
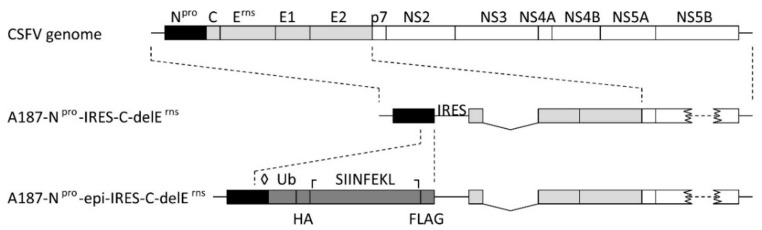
Design of the backbone replicon A187-N^pro^-epi-IRES-C-delE^rns^. The map of the backbone replicon (packaged in VRP 0) used in this study is represented schematically, with the N^pro^ gene shown in black, the structural protein sequences in light gray, and the non-structural protein region in white. The cassette for ubiquitinated and tagged epitope expression is shown in dark gray and encodes a non-cleavable ubiquitin (Ub), a haemaglutinin tag (HA), an epitope site, and a FLAG tag. Additionally, a C_138_A mutation was introduced in N^pro^ to destroy the N^pro^-mediated interferon regulatory factor 3 degradation (◊), and the sequence coding for the SIINFEKL control epitope was flanked by an upstream *Kas*I restriction site (┌) and a downstream *Mlu*I restriction site (┐) for replacement of the SIINFEKL sequence with the porcine reproductive and respiratory syndrome virus (PRRSV) polyepitope sequences.

**Figure 3 vaccines-09-00208-f003:**
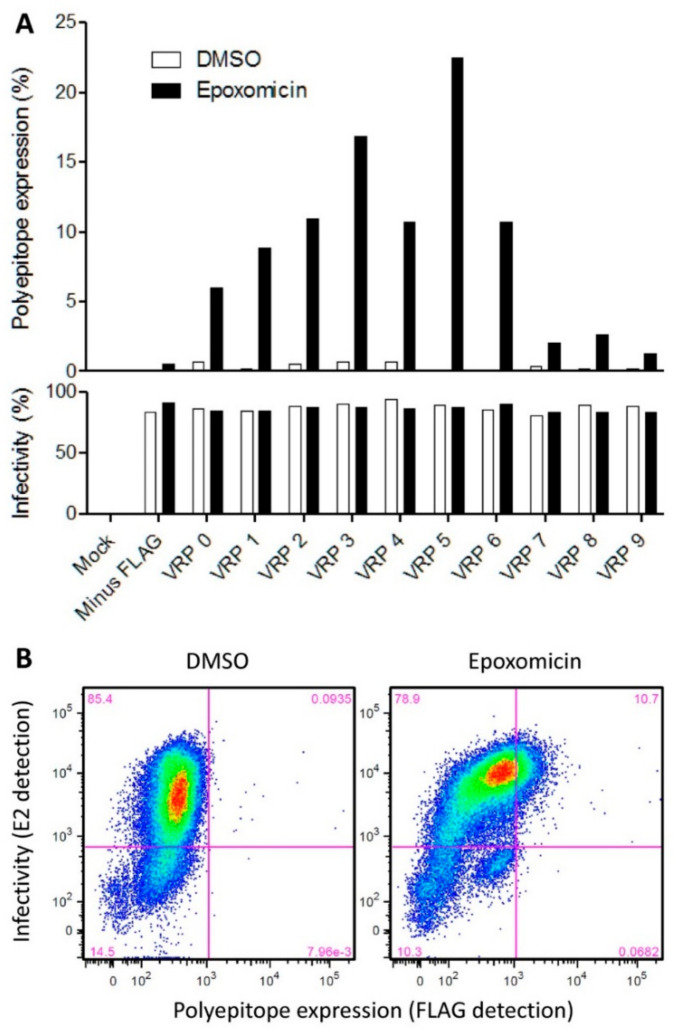
Verification of VRP infectivity and polyepitope expression in SK-6 cells. (**A**) 10^5^ cells were infected at a MOI of 5 with the VRPs 0 to 9 or the VRP lacking FLAG (A187-N^pro^-IRES-C-delE^rns^) or mock-infected as control. Twenty-eight hour post-infection cells were treated with the proteasome inhibitor epoxomicin (100 nM) or dimethyl sulfoxide (DMSO). After an additional 17.5 h, the percentage of cells expressing the polyepitope (top panel) and the percentage of infected cells (bottom panel) were determined by FCM using anti-FLAG and anti-E2 antibodies, respectively. At least 36,000 events were acquired for each sample. (**B**) The FCM plots of VRP 6-infected cells under DMSO *versus* epoxomicin treatment are shown as an example.

**Figure 4 vaccines-09-00208-f004:**
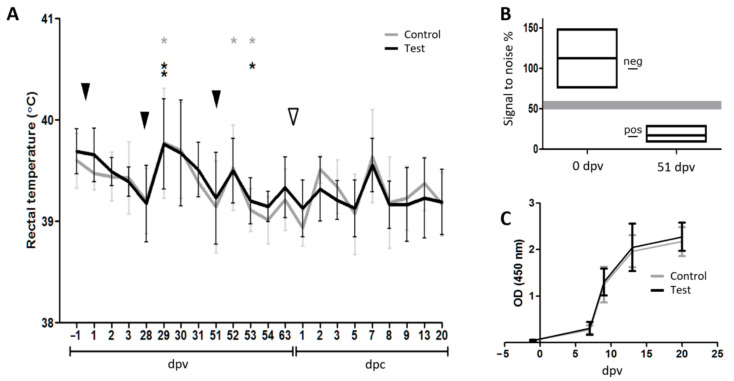
The vaccination challenge trial. (**A**) Rectal temperature throughout the experiment. The black and grey lines represent the average temperatures of the test group and the control group, respectively. Error bars indicate the standard deviations. The black arrowheads indicate the vaccination events. The white arrowhead indicates the challenge of the pigs with PRRSV-2. Before challenge, asterisks indicate group-wise temperature peaks that are significantly different from the previous day calculated using a two-tailed paired Student’s *T*-test (*: 0.05 ≥ *p* > 0.01; **: 0.01 ≥ *p* > 0.001). After challenge, no significant differences in body temperature were observed between the two groups as calculated using a two-tailed unpaired Student’s *T*-test. (**B**) Seroconversion following vaccination. Box diagram of the anti-E2 inhibition ELISA performed at dpv 0 and dpv 51 illustrating that the pigs responded to the vaccine vector by producing antibodies against the classical swine fever virus (CSFV) protein E2. The upper and lower edges of the boxes represent maximum and minimum signal-to-noise percentage values, respectively. The mean values are represented by the middle lines. The positive (11, 85) and negative (100) control values are indicated by lines in the space between the two boxes. The shaded area represents the threshold range. Note that no data points were observed in this area. (**C**) Seroconversion to PRRSV following challenge infection. Line diagram of the PRRSV Ab ELISA showing that all pigs responded to the challenge by producing antibodies against PRRSV N.

**Figure 5 vaccines-09-00208-f005:**
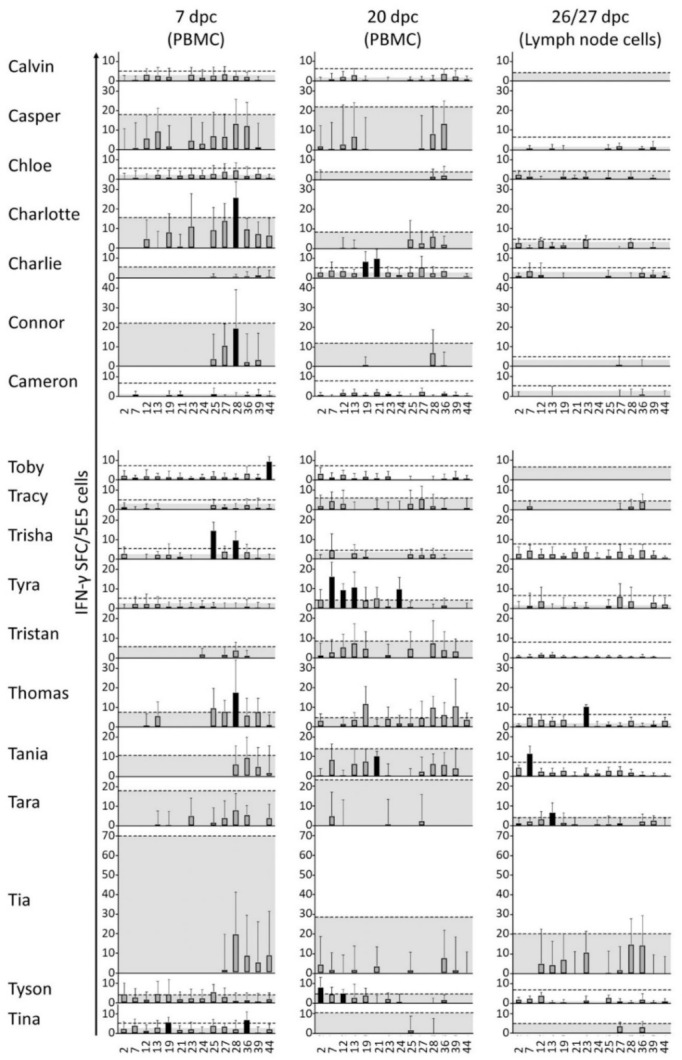
Overview of cell-mediated immune responses to peptides post-challenge. At dpc 7 and 20, 5 × 10^5^ freshly purified peripheral blood mononuclear cells (PBMCs) were restimulated separately with 14 selected peptides and with media as a background control. At euthanasia (dpc 26/27), 3 × 10^5^ cells derived from the lymph node were treated the same way and the counts were normalized to 5 × 10^5^ cells/well for comparison with the PBMC counts. The response to restimulation is presented here as columns indicating the average number of IFN-γ spot forming cells (SFC) in response to restimulation with peptide (signal) minus the average number of SFC in response to restimulation with medium (background). Error bars represent the corresponding standard deviations. Upper edges of the gray area represent the subtracted average backgrounds for reference. Dashed lines indicate ratio-2 thresholds. This is defined either as 2 × background, or as 8 representing the limit of detection in cases where 2 × background is less than 8. Black columns represent positive peptide responses according to the DFR method. Peptides are indicated on the x-axis with reference to the ID column in Table 2.

**Figure 6 vaccines-09-00208-f006:**
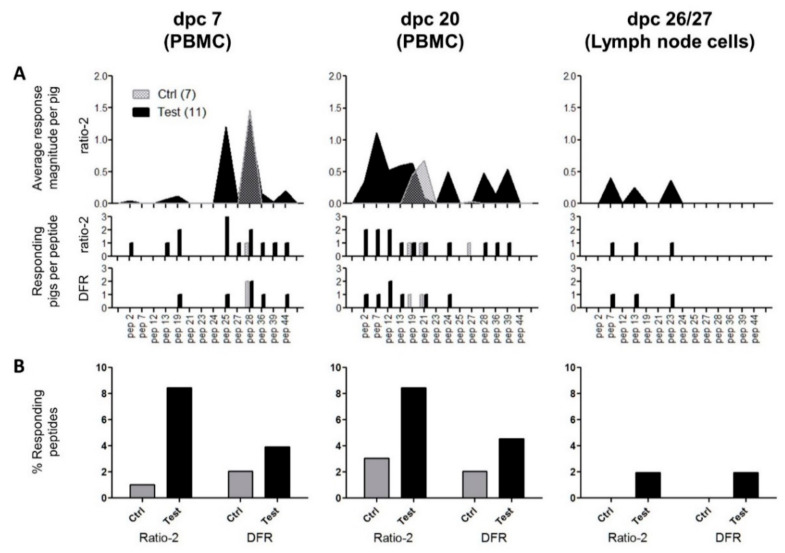
Summary of cell-mediated immune responses to peptides post-challenge. (**A**) The top panel represents the average magnitude of significant positive responses per pig according to the ratio-2 method; the middle panel represents the number of pigs/group with a significant positive response to the peptides according to the ratio-2 method; and the bottom panel represents the number of pigs/group with a significant positive response to the peptides according to the DFR method. Gray fill: control group (N = 7); black fill: test group (N = 11). The peptides are indicated on the x-axis with reference to the ID column in Table 2. (**B**) Relative comparison of responding peptides according to date, group, and statistical method. Each column was calculated using the formula: (responding peptides)/(tested peptides)/(number of pigs in the specific group).

**Figure 7 vaccines-09-00208-f007:**
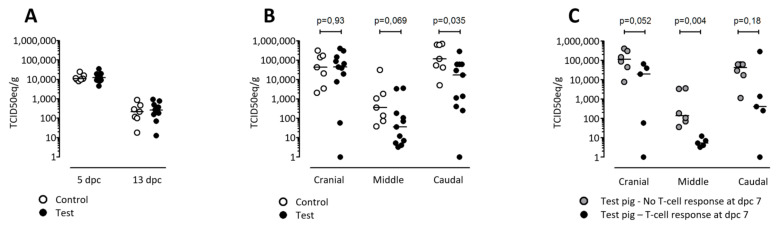
Virus load in serum and lung tissue analyzed by qRT-PCR. (**A**): Virus load in the serum at dpc 5 and 13 given in TCID_50_ equivalents per ml serum. (**B**): Virus load in the lung tissue from cranial, middle, and caudal parts of the lung prepared from cutouts of 0.2–0.4 g and normalized to TCID_50_ equivalents per 1 g of tissue. (**C**): Virus load in lung tissue from test pigs only, displayed as non-T-cell-responders *versus* T-cell-responders at dpc 7, differentiated using the ratio-2 method. All P-values were calculated using Mann–Whitney. All measurements were performed in duplicates and were converted from cq-values using a standard curve based on a purified 10-fold dilution series of the challenge isolate. Group medians are indicated with a line. Virus load in the lung tissue from cranial, middle and caudal parts of the lung were prepared from cutouts of 0.2–0.4 g and normalized to TCID_50_ equivalents to 1 g of tissue.

**Table 1 vaccines-09-00208-t001:** Background of the 18 pigs included in the vaccine-challenge experiment.

Group	Pig	SLA-1*04:01	SLA-1*07:02	SLA-2*04:01	Weight (kg)	Litter	Pen
Control	Calvin	●	●		8	2	2
	Casper	●	●		9.6	2	1
	*Chloe*	●	●		11.7	3	2
	*Charlotte*	●	●		5.7	4	2
	Charlie	●		●	8.5	1	1
	Connor	●		●	7.1	1	1
	Cameron	●		●	8.5	3	1
Test	Toby	●	●		8.9	1	1
	*Tracy*	●	●		8.1	2	2
	*Trisha*	●	●		9.2	2	1
	*Tyra*	●	●		10.3	2	1
	Tristan	●	●		6.1	3	2
	Thomas	●	●		7.8	4	2
	*Tania*	●		●	7.4	1	2
	*Tara*	●		●	6.7	1	1
	*Tia*	●		●	5.7	1	1
	Tyson	●		●	10.2	1	2
	*Tina*	●		●	10.8	3	2

Pigs were distributed between the control group (N = 7) and test group (N = 11) for an even distribution of swine leukocyte antigens (SLA) profile, body weight, and litter. All pigs were kept in the same pen during the first seven weeks after which they were split in two neighboring pens due to space constraints. Female pigs are in italic.

**Table 2 vaccines-09-00208-t002:** Peptides included in the VRPs, in post-challenge ELISPOT, and naturally expressed by the challenge strain.

ID	Sequence	SLA-1*04:01			SLA-1*07:02			SLA-2*04:01			In chal. Strain	In post-chal. ELISPOT
		In VRP	Stab. (h)	Aff. (nM)	In VRP	Stab. (h)	Aff. (nM)	In VRP	Stab. (h)	Aff. (nM)		
2	YAQHMVLSY		−	−	●	0.9	4	●	1.1	60	√	√
4	YSFPGPPFF		0.2	37 †	●	0.2	18,708		−	−		
5	RALPFTLSNY				●	0.3	12		0.1	−	√	
7	QVYERGCRWY	●	0.3	682	●	4.5	168	●	0.7	209 †	√	√
9	IVYSDDLVLY		−	−	●	0.5	13		−	−	√	
10	KVAHNLGFYF	●	0.3	122	●	0.3	99				√	
11	TRARHAIFVY				●	0.3	60		0.2	−	√	
12	LSFSYTAQF							●	1.3	73	√	√
13	FTWYQLASY		0.1	92 †	●	0.2	62	●	9.1	2	√	√
17	RTAIGTPVY	●	0.5	57	●	0.2	1852	●	−	385 †	√	
18	YTAQFHPEIF		−	−	●	0.2	24,378	●	0.4	−	√	
19	LSDSGRISY	●	1.1	10	●	0.2	383		0.2	4182 †	√	√
21	KVAHNLGFY	●	1.5	4	●	2.8	11		−	−	√	√
22	KIFRFGSHKW	●	0.2	98					0.1	9 †	√	
23	NISAVFQTYY	●	0.1	413	●	0.9	6	●	−	862	√	√
24	RTAPNEIAF	●	2.1	4					0.1	−		√
25	ASDWFAPRY	●	4.9	2	●	0.2	71		−	−	√	√
27	RPFFSSWLV				●	37.4	1				√	√
28	FVLSWLTPW		−	−	●	0.2	1372	●	13.7	3	√	√
29	MVNTTRVTY		0.1	206 †	●	0.2	47		−	−	√	
30	CVFFLLWRM				●	0.2	283				√	
33	ITANVTDENY		0.1	−	●	0.3	69		−	−	√	
34	SSEGHLTSVY		−	−	●	0.2	12,701 †		0.1	1692 †	√	
36	LTAALNRNRW		−	−				●	3.6	40	√	√
38	LSASSQTEY		0.1	91 †	●	0.2	479		0.2	−	√	
39	VRWFAANLLY				●	2.7	44				√	√
43	TTMPSGFELY	●	−	576	●	0.8	6		0.2	1838 †		
44	MSWRYSCTRY		−	−	●	0.5	87	●	1.5	15		√
46	ALATAPDGTY		−	607 †	●	0.1	2736				√	
48	WGVYSAIETW							●	0.2	21,433	√	
49	FLNCAFTFGY		−	−	●	0.3	20	●	0.3	2069	√	
50	NSFLDEAAY				●	0.1	43		−	−	√	
53	MPNYHWWVEH				●	0.6	32				√	

Data on the 33 epitopes used in the polyepitopes. Provided information with regard to the three SLAs: Inclusion (●) of epitope in SLA-specific VRP, measured binding stability (average dissociation half-life in decimal hour (h)), and measured binding affinity (average equilibrium dissociation constant (nM)). †: only one successful affinity measurement was obtained. Hyphen (-): no successful measurements were obtained (stability or affinity). Empty field: not tested.

**Table 3 vaccines-09-00208-t003:** Polyepitopes assembled according to the SLA specificity of the peptides.

VRP	SLA Specificity	Epitopes Succession in Polyepitope	Epitopes
VRP 1	SLA-1*04:01	19-23-43-25-24-17-7-22-10	9
VRP 2	SLA-1*04:01	23-7-22-10-25-24-43-19-17	9
VRP 3	SLA-1*04:01	19-17-7-22-10-43-25-24-23	9
VRP 4	SLA-1*07:02	34-33-10-5-18-17-13-29-46-43-2-50-28-39-19-30-11-25-23-4-49-53-7-27-44-38-9	27
VRP 5	SLA-1*07:02	13-18-38-27-23-11-25-19-7-53-46-2-17-4-33-49-9-39-5-28-10-29-44-30-34-50-43	27
VRP 6	SLA-1*07:02	2-13-34-33-18-23-50-53-5-28-39-7-29-46-17-9-49-25-4-30-27-43-11-19-44-38-10	27
VRP 7	SLA-2*04:01	23-13-44-7-2-48-28-12-18-36-17-49	12
VRP 8	SLA-2*04:01	2-12-23-44-7-18-13-48-36-17-49-28	12
VRP 9	SLA-2*04:01	13-12-28-2-44-7-48-36-17-18-23-49	12

Overview of the pseudo-triplicate polyepitopes for the formation of the VRPs 1–9. Epitope numbers refer to ID column in Table 2. SLA specificity and number of epitopes for reference.

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
