# Peer review of "Reduced Virus Load in Lungs of Pigs Challenged with Porcine Reproductive and Respiratory Syndrome Virus after Vaccination with Virus Replicon Particles Encoding Conserved PRRSV Cytotoxic T-Cell Epitopes"

_vaccines, 2021, doi:10.3390/vaccines9030208_

Round 1

Reviewer 1 Report

In the manuscript of Welner et al., the authors describe the generation of virus replicon particles (VRP) based on a classical swine fever virus genome incapable of producing infectious progeny and designed to express conserved PRRSV-2 cytotoxic T-cell epitopes. The vaccine was tested in pigs matched with the epitopes by their swine leucocyte antigen profiles. Vaccine induced CTL immune responses in pigs. Challenged PRRSV strain was too mild, and did not cause any clinical signs of disease. Challenge virus load in serum did not differ significantly between the groups of pigs, whereas the virus load in the caudal part of the lung was significantly lower in the test group compared to the control group. Overall, this research contributes to the development of improved vaccine against PRRSV and to better understanding the role of cell-mediated immune response in control of the viral infection.

Minor suggestions:

Line 52 - change capital V in the word “Vaccination” to small one.

Lines 50-56 - in this paragraph the following may be added, “MLV is not approved for use in PRRSV-negative herds and breeding age boars.”

Lines 201-212 – mention how many times pigs were vaccinated and time of vaccination and challenge; figure 3A may be referenced here.

Line 244 – change “in a microscope” to “under a microscope”

Line 345 – peptides was not included in the challenge strain, they are naturally present in it

Line 550 – mistake in text formatting; figure legend must be under the figure.

General comment: I have never seen a study, where pigs were given personnel names, not just the numbers.  

Author Response

In the manuscript of Welner et al., the authors describe the generation of virus replicon particles (VRP) based on a classical swine fever virus genome incapable of producing infectious progeny and designed to express conserved PRRSV-2 cytotoxic T-cell epitopes. The vaccine was tested in pigs matched with the epitopes by their swine leucocyte antigen profiles. Vaccine induced CTL immune responses in pigs. Challenged PRRSV strain was too mild, and did not cause any clinical signs of disease. Challenge virus load in serum did not differ significantly between the groups of pigs, whereas the virus load in the caudal part of the lung was significantly lower in the test group compared to the control group. Overall, this research contributes to the development of improved vaccine against PRRSV and to better understanding the role of cell-mediated immune response in control of the viral infection.

Minor suggestions:

Line 52 - change capital V in the word “Vaccination” to small one.

Answer and modification: Corrected (line 53)

Lines 50-56 - in this paragraph the following may be added, “MLV is not approved for use in PRRSV-negative herds and breeding age boars.”

Answer and modification: The paragraph has been expanded to include the suggested comment (lines 57-62)

Lines 201-212 – mention how many times pigs were vaccinated and time of vaccination and challenge; figure 3A may be referenced here.

Answer and modification: number of vaccinations and intervals between stated in lines 234-236. Time of challenge stated in line 238-9. Reference to figure 1 (old figure 3A) in line 239.

Line 244 – change “in a microscope” to “under a microscope”

Answer and modification: Corrected (line 274)

Line 345 – peptides was not included in the challenge strain, they are naturally present in it

Answer and modification: Rephrasing of table titel (line 378)

Line 550 – mistake in text formatting; figure legend must be under the figure.

General comment: I have never seen a study, where pigs were given personnel names, not just the numbers.  

Answer: These names were assigned after the end of the experiment for a better distinction of control and test pigs. To maintain blinding in the execution of the experiment, the pigs were only identified by ear tag numbers during the experiment. Modification: We clarified this and explained the names on lines 216-218

Reviewer 2 Report

In this manuscript, the authors used a non-cytopathic CSFV VRP to express 9 different polyepitopes resulting from different combinations of 33 conserved PRRSV-2 T cell epitopes that have binding affinity and stability to three MHC class I swine leukocyte antigens (SLA). The resultant VRPs were rescued and in vitro characterized. Then, pigs with the corresponding three SLAs were vaccinated with VRPs followed by challenge with a PRRSV-2 strain to evaluate the protective efficacy. It was found VRP vaccination had no effect on the prevention of infection or reduction of viremia compared to the mock vaccination group, although the authors stated that vaccination could reduce virus load in the lungs. The results are disappointing. However, it is difficult and complicated to study PRRSV CTL epitopes and T cell immune responses. The authors are applauded for their efforts in this research area. The data is still a valuable addition to the archives of PRRSV T cell epitopes and cell-mediated immunity. Nevertheless, the manuscript needs to be improved in various areas.

The organization of the manuscript. Some relevant information was not described clearly in Materials and Methods; instead, such information was described in Result section. It does not flow well. Some examples are given below.

  1. Result 3.1.1, 3.1.2, 3.2.1, Table 1, Table 2, and Figure 1 will be more appropriate in M&M.
  2. Pig experimental design was poorly described and explained in M&M but was described in detail in Result section.

M&M 2.6. According to “19 randomly sampled weaners”, it appears that 19 pigs were used.  However, in the Result section, it appears that 18 pigs were used. For sentence “After seven weeks of acclimatization, the pigs were separated in two pens…”. Should it be “After seven days of acclimatization”?

According to 3.2.2, a low pathogenicity of PRRSV strain was used as the challenge virus. Generally speaking, virulent virus strains are selected as challenge virus to evaluate protective efficacy. It is unclear why the authors chose a low pathogenic PRRSV strain as a challenge virus in this study.  Please explain and justify it.

In M&M 2.12 IFN-gamma ELISOPT, the authors mentioned that stimulation was not only induced with individual peptides but also induced with the VRP mixture used for vaccination and the PRRSV strain used for challenge. In Result section, the IFN-gamma ELISPOT data related to stimulation with VRP mixture and PRRSV challenge strain were not presented. Please clarify the data.

The authors concluded that the pigs of the test group have a generally higher CMI response frequency and magnitude than the pigs of the control group. However, in Figure 4, it is pretty evident that there were big variations of the CMI response between pigs regardless of test group or control group.  This should be clearly pointed out.

In Discussion, the authors are recommended to give some take-home messages from the current study. What researchers and swine practitioners can learn about PRRSV CTL epitopes and the potential problem of using this approach for vaccine development?  Even though all pigs used in this study were selected to have SLAs reacting with the VRP PRRSV-2 epitopes, big variations of inducing CMI responses were still observed between pigs. How to address this issue when developing CTL epitopes-based PRRSV-2 vaccines?  In addition, in the real world, pigs may have lots of SLAs, how to develop more general and uniform CTL epitopes-based PRRSV vaccines for use in most pig populations? 

Author Response

In this manuscript, the authors used a non-cytopathic CSFV VRP to express 9 different polyepitopes resulting from different combinations of 33 conserved PRRSV-2 T cell epitopes that have binding affinity and stability to three MHC class I swine leukocyte antigens (SLA). The resultant VRPs were rescued and in vitro characterized. Then, pigs with the corresponding three SLAs were vaccinated with VRPs followed by challenge with a PRRSV-2 strain to evaluate the protective efficacy. It was found VRP vaccination had no effect on the prevention of infection or reduction of viremia compared to the mock vaccination group, although the authors stated that vaccination could reduce virus load in the lungs. The results are disappointing. However, it is difficult and complicated to study PRRSV CTL epitopes and T cell immune responses. The authors are applauded for their efforts in this research area. The data is still a valuable addition to the archives of PRRSV T cell epitopes and cell-mediated immunity. Nevertheless, the manuscript needs to be improved in various areas.

The organization of the manuscript. Some relevant information was not described clearly in Materials and Methods; instead, such information was described in Result section. It does not flow well. Some examples are given below.

Result 3.1.1, 3.1.2, 3.2.1, Table 1, Table 2, and Figure 1 will be more appropriate in M&M.

Pig experimental design was poorly described and explained in M&M but was described in detail in Result section.

Answer and modifications: Regarding section 3.2.1, this section has now been deleted from the results section and all information regarding the experimental design has been transferred to section 2.6 and 2.7. Consequently, previously named table 3 and figure 3A were moved and renamed to table 1 and figure 1, respectively. Regarding section 3.1.1, section 3.1.2, table 1, table 2, and figure 1 (now renumbered to table 2, table 3 and figure 2, respectively) the reviewer’s concern has previously been a subject of debate within the authors, but we have agreed to use the current outline. In our study, we have a 3-step approach: M&M -> PRODUCTION RESULTS (product from design and construction procedure) -> INFECTION RESULTS (from animal experiment). This issue arises as we attempt to describe this 3-step approach within a 2-step template, so obviously the decision lies in whether ‘PRODUCTION RESULTS’ should belong to M&M or to Results. There are good arguments for both, and at the end of the day, it is merely a question of the eyes that see.

M&M 2.6. According to “19 randomly sampled weaners”, it appears that 19 pigs were used.  However, in the Result section, it appears that 18 pigs were used. For sentence “After seven weeks of acclimatization, the pigs were separated in two pens…”. Should it be “After seven days of acclimatization”?

Answer and modification: The 19 randomly sampled weaners were tested for PRRSV as an extra precaution to ensure that the herd was in fact PRRSV-free before piglets were bought. The 18 experimental animals were born afterwards, but came from the same herd. It has been rephrased to avoid confusion in lines 207-208. Regarding ‘After seven weeks of acclimatization, the pigs were separated in two pens…’, this is correct, but has been rephrased in lines 218-219. Confusion may arise since the first vaccination was given after seven days – not weeks. This has been rephrased in line 235 (which was part of section 3.2.1 that has now been integrated into section 2.6 and 2.7.

According to 3.2.2, a low pathogenicity of PRRSV strain was used as the challenge virus. Generally speaking, virulent virus strains are selected as challenge virus to evaluate protective efficacy. It is unclear why the authors chose a low pathogenic PRRSV strain as a challenge virus in this study.  Please explain and justify it.

Answer and modification: The challenge strain used was a field strain belonging to the linkage 5.2 which is the linkage circulating in Europe. We used this for challenge since we have previously used this for exp studies because this strain is relevant for the European field situation. The sentence in section 3.2 has been rephrased (lines 457-458) and the impact of the challenge strain on the outcome of the study has now been addressed in the discussion section (lines 736-742).

In M&M 2.12 IFN-gamma ELISOPT, the authors mentioned that stimulation was not only induced with individual peptides but also induced with the VRP mixture used for vaccination and the PRRSV strain used for challenge. In Result section, the IFN-gamma ELISPOT data related to stimulation with VRP mixture and PRRSV challenge strain were not presented. Please clarify the data.

Answer: Restimulation with virus and VRP were excluded from the setup due to limitations in test capacity and PBMCs available. Modification: Briefly explained in M&M lines 330-331, and elaborated on in Results lines 496-503

The authors concluded that the pigs of the test group have a generally higher CMI response frequency and magnitude than the pigs of the control group. However, in Figure 4, it is pretty evident that there were big variations of the CMI response between pigs regardless of test group or control group.  This should be clearly pointed out.

Answer and modification: Specified in line 655 that the T-cell response readouts were highly variable and of low magnitude.

In Discussion, the authors are recommended to give some take-home messages from the current study. What researchers and swine practitioners can learn about PRRSV CTL epitopes and the potential problem of using this approach for vaccine development?

Answer and modification: New paragraph addressing this in lines 713-732

Even though all pigs used in this study were selected to have SLAs reacting with the VRP PRRSV-2 epitopes, big variations of inducing CMI responses were still observed between pigs. How to address this issue when developing CTL epitopes-based PRRSV-2 vaccines? 

Answer and modification: New paragraph addressing this in lines 713-732

In addition, in the real world, pigs may have lots of SLAs, how to develop more general and uniform CTL epitopes-based PRRSV vaccines for use in most pig populations? 

Answer and modification: New paragraph addressing this in lines 713-732

Reviewer 3 Report

This manuscript describes a novel and well-designed approach to elicit cross-protective T cell responses against PRRSV-2.

Overall, the manuscript is well written and presented.

Given the approach, it is highly unfortunate that T cell data could not be interpreted pre-challenge nor was there any attempt to assess CD8 versus CD4 T cell responses.

I find it challenging to reconcile the very weak peptide specific responses observed post-challenge with the reduced viral loads in the lungs of challenged pigs (Did the authors consider assessing responses of T cells isolated from bronchoalveolar lavage?). Nevertheless, there is an effect on viral loads in the lungs which supports the further investigation of this approach.

Nevertheless, I think this manuscript will be of interest to the field and I think it is suitable for publication subject to the authors addressing the points below.

Abstract:

  • I suggest this could be phrased better: “At most, control of PRRSV is maintained by the use of modified live virus vaccines, which is accompanied by multiple safety issues”.
  • Make it clearer these IFN-g responses are post-challenge.

Introduction

  • Line 39-40: Rephrase “…or a hemorrhagic ‘Porcine High Fever Disease’ with virulent
  • strains from South-East Asia [3] and the US [4].”
  • Line 40: USA
  • Line 59-60: viral vectors should also be mentioned.
  • Line 63-64 and elsewhere: Rather than “cell mediated immunity (CMI)” I think it is more appropriate to refer to T cell responses.
  • Line 69-71: Rephrase this. Passive transfer experiments have shown that antibodies alone can provide (complete) protection.
  • Line 73: Viral vectors
  • Line 74: “heterologous antigens” or “vaccine candidate antigens” rather than “proteins of interest”.
  • Line 96-97: For the benefit of the more general reader, I think it would be beneficial to expand on choice of CSFV as the replicon system, e.g., the natural cellular tropism of CSFV.

Materials and Methods

  • Line 153: “APHA” not “AHVLA”
  • Line 169-170: Consider rephrasing “poly-ubiquitinylated epitopes” to “ubiquitinated poly-epitopes”.
  • Line 190: Remove “The”
  • Line 191: Remove “The” from the the start of the sentence. Describe the sex and breed of the piglets.
  • Line 205-207: State how many times pigs were vaccinated and the interval between.
  • Line 209: State when post-vaccination pigs were challenged.
  • Line 223: Remove “porcine”.
  • Line 226: State centrifugal speed as x g not rpm.
  • Line 236: State when pigs were euthanised (dpc).
  • Line 282: I may have missed this, but the supplier of the peptides and the solvent used to dissolve peptides should be described.

Results

  • Line 333 and 346 (and possibly elsewhere): I think at this stage it is more appropriate to refer to “predicted epitopes” rather than “epitopes”.
  • Figure 2 - appears in the manuscript before any reference is made to it in the text.
  • Line 437-438: Reference is made to “significant variations” does this refer to statistically significant differences?
  • Line 466-468: Does seroconversion to CSFV E2 really indicate that “VRP-encoded genes were efficiently transcribed and translated into protein in the vaccinated animals”? Isn’t E2 expressed on the surface of CSFV VRP?
  • Line 437: T cell responses rather than CMI
  • Line 474-477: The pre-challenge IFN-g ELISPOT data should be included as supplementary data.
  • Line 479-480: It should be clarified why 14 individual peptides were used for the post-challenge ELISPOT rather than the 12 matrix pools that encompassed all peptides. It is also a pity that PRRSV (or VRPs) were not included in these assays to benchmark the peptide responses.
  • Line 485: “the DFR method allows a better statistical comparison…”. It is not clear from the subsequent test what statistical analyses were performed and whether significant differences in responses were observed?
  • Line 497: “serine”
  • Line 499: “alanine”
  • Line 544: Rephrase “duration of viremia” since this does not appear to have been assessed.
  • Could the authors make more effort to link peptide responses to the haplotypes of the individual animals.

Discussion

  • Line 574: Because VRPs are replication defective is it relevant/necessary to state “non-cytopathogenic”?
  • Line 591: I think the abbreviation CTL has been introduced/used before so it should be used here too.
  • Line 648: I am a little confused by this statement “Immunoblot analysis of lysates of epoxomicin-treated VRP-infected cells in vitro failed twice to show visible bands (data not shown)”. Presumably this was a complimentary analysis performed alongside the flow cytometry? I agree the flow cytometry data also suggest very low-level expression of most of the polyepitopes.
  • Line 664-672: I would recommend removing or significantly revising this paragraph as it is highly speculative. If strong responses were induced by vaccination, they should still be observable above a background.

Author Response

This manuscript describes a novel and well-designed approach to elicit cross-protective T cell responses against PRRSV-2. Overall, the manuscript is well written and presented. Given the approach, it is highly unfortunate that T cell data could not be interpreted pre-challenge nor was there any attempt to assess CD8 versus CD4 T cell responses.

I find it challenging to reconcile the very weak peptide specific responses observed post-challenge with the reduced viral loads in the lungs of challenged pigs (Did the authors consider assessing responses of T cells isolated from bronchoalveolar lavage?). Nevertheless, there is an effect on viral loads in the lungs which supports the further investigation of this approach.

Nevertheless, I think this manuscript will be of interest to the field and I think it is suitable for publication subject to the authors addressing the points below.

Answer and modification: We agree with the reviewer’s concerns and have elaborated on this in lines 689-694.

Abstract:

I suggest this could be phrased better: “At most, control of PRRSV is maintained by the use of modified live virus vaccines, which is accompanied by multiple safety issues”.

Answer and modification: Rephrased (lines 17-19)

Make it clearer these IFN-g responses are post-challenge.

Answer and modification: “…after challenge…” included in sentence (line 28)

Introduction:

Line 39-40: Rephrase “…or a hemorrhagic ‘Porcine High Fever Disease’ with virulent

strains from South-East Asia [3] and the US [4].”

Answer and modification: Rephrased (lines 39-41)

Line 40: USA

Answer and modification: corrected (line 41)

Line 59-60: viral vectors should also be mentioned.

Answer and modification: Viral vectors now mentioned (line 63)

Line 63-64 and elsewhere: Rather than “cell mediated immunity (CMI)” I think it is more appropriate to refer to T cell responses.

Answer and modification: The authors agree on this observation, and it has been corrected throughout the manuscript

Line 69-71: Rephrase this. Passive transfer experiments have shown that antibodies alone can provide (complete) protection.

Answer and modification: This section has been rephrased to include an article providing evidence of the induction of sterilizing immunity without CMI (ref 19), and an article reporting protective immunity without the presence of neutralizing Abs (ref 20) lines 74-80)

Line 73: Viral vectors.

Answer and modification: Corrected (line 82)

Line 74: “heterologous antigens” or “vaccine candidate antigens” rather than “proteins of interest”.

Answer and modification: rephrased (line 83)

Line 96-97: For the benefit of the more general reader, I think it would be beneficial to expand on choice of CSFV as the replicon system, e.g., the natural cellular tropism of CSFV.

Answer and modification: We agree with the reviewer and elaborated on this with 4 additional references (lines 111-115)

Materials and Methods:

Line 153: “APHA” not “AHVLA”.

Answer and modification: Corrected (line 166)

Line 169-170: Consider rephrasing “poly-ubiquitinylated epitopes” to “ubiquitinated poly-epitopes”.

Answer and modification: Corrected (line 182-183)

Line 190: Remove “The”

Answer and modification: Corrected (line 204)

Line 191: Remove “The” from the the start of the sentence. Describe the sex and breed of the piglets.

Answer and modification: The breed is described in line 210. Sex is described in line 211.

Line 205-207: State how many times pigs were vaccinated and the interval between.

Answer and modification: number of vaccinations and intervals between stated in lines 234-236.

Line 209: State when post-vaccination pigs were challenged.

Answer and modification: Time of challenge stated in line 238-9.

Line 223: Remove “porcine”.

Answer and modification: Corrected (line 253)

Line 226: State centrifugal speed as x not rpm.

Answer and modification: Corrected (line 256)

Line 236: State when pigs were euthanised (dpc).

Answer and modification: Corrected (line 266)

Line 282: I may have missed this, but the supplier of the peptides and the solvent used to dissolve peptides should be described.

Answer: The reviewer did not miss it – we did. modification: Supplier of peptides stated in line 294 and description of peptide dissolution and storage described in new supplementary data 2, being referred to in line 295.

Results:

Line 333 and 346 (and possibly elsewhere): I think at this stage it is more appropriate to refer to “predicted epitopes” rather than “epitopes”.

Answer: We apologize if it was not sufficiently clear that the 33 peptides were previously verified as binders in vitro. We acknowledge that this is not the same as being biologically effective as epitopes, but it is still a closer approximation to their biological role than if they were only selected based on an in silico prediction. The rationale for selecting the 33 epitopes is first mentioned in the introduction in lines 108-111, but for the sake of clarity we have specified this further in other sections. Modification: In section 2.1 the in vitro verification of the epitopes binding capacity is mentioned (lines 123-124). In section 2.6 it is specified that complex formation of the predicted epitopes was demonstrated experimentally in a previous study (lines 204-5).

Figure 2 - appears in the manuscript before any reference is made to it in the text.

Answer and modification: This may be an artifact of track changes or issues with compatibility. The issue was not present in the version that we originally submitted, but was there when we received the revision. We have now corrected it, but expect that it might happen again. We assume that the typesetter will correct this before publication.

Line 437-438: Reference is made to “significant variations” does this refer to statistically significant differences?

Answer: Yes, these significant variations are statistically significant and are calculated as described in section 2.14. Modification: ‘Statistically’ inserted in line 455

Line 466-468: Does seroconversion to CSFV E2 really indicate that “VRP-encoded genes were efficiently transcribed and translated into protein in the vaccinated animals”? Isn’t E2 expressed on the surface of CSFV VRP?

Answer and modification: We elaborated on this on lines 482-6 of the revised manuscript, referring to data from Suter et al demonstrating that UV-inactivated VRP do not induce seroconversion against E2 of the incoming VRP envelope.

Line 437: T cell responses rather than CMI.

Answer and modification: Corrected (line 491)

Line 474-477: The pre-challenge IFN-g ELISPOT data should be included as supplementary data.

Answer and modification: The quality of the pre-challenge ELISPOT was highly inconsistent between individual wells, and despite numerous attempts to digitalize data (either by use of a spot counter or under a microscope) all attempts were abandoned as it showed impossible to extract anything but noise. Consequently, the only data we have from the pre-challenge experiment is several hundreds of un-analyzed photos of the individual wells. These could in theory be submitted as supplementary data, but we see no point in this.

Line 479-480: It should be clarified why 14 individual peptides were used for the post-challenge ELISPOT rather than the 12 matrix pools that encompassed all peptides. It is also a pity that PRRSV (or VRPs) were not included in these assays to benchmark the peptide responses.

Answer: Restimulation with virus and VRP were excluded from the setup due to limitations in test capacity and obtainable PBMCs. This was the trade-off for increasing the sensitivity by increasing number of multiplicates and cells per well. For the same reasons only single epitopes were analyzed in the post-challenge ELISPOT. Modification: Briefly explained in M&M lines 330-331, and elaborated on in Results lines 495-503

Line 485: “the DFR method allows a better statistical comparison…”. It is not clear from the subsequent test what statistical analyses were performed and whether significant differences in responses were observed?

Answer and modification: Rephrasing of section in line 504-9 to clarify.

Line 497: “serine”

Answer and modification: Corrected (line 521)

Line 499: “alanine”

Answer and modification: Corrected (line 523)

Line 544: Rephrase “duration of viremia” since this does not appear to have been assessed.

Answer and modification: Duration (of viremia) deleted at line 568, since viral load in serum was only measured at dpc 5 and 13

Could the authors make more effort to link peptide responses to the haplotypes of the individual animals.

Answer and modification: We agree that this would be interesting and optimal, but given the fact that we are only certain of 2 out of up to 6 expressed SLA alleles per animal, it would be mere speculation, so we have decided not to pursue this any further.

Discussion:

Line 574: Because VRPs are replication defective is it relevant/necessary to state “non-cytopathogenic”?

Answer: Replication-defective relates to the incapacity of the replicon to produce infectious progeny virus. But the CSFV RNA replicates without resulting in cell death, which is different from alphavirus or VSV-derived replicons for instance. Therefore, it is essential to mention that the replicons are non-cytopathogenic. Modification: We removed ”non-cytopathogenic” in the discussion (line 597), but maintained it in the introduction (line 106). In addition we wrote ”RNA replication” instead of ”replication” on line 83 to avoid any confusion.

Line 591: I think the abbreviation CTL has been introduced/used before so it should be used here too.

Answer and modification: Corrected (line 615)

Line 648: I am a little confused by this statement “Immunoblot analysis of lysates of epoxomicin-treated VRP-infected cells in vitro failed twice to show visible bands (data not shown)”. Presumably this was a complimentary analysis performed alongside the flow cytometry? I agree the flow cytometry data also suggest very low-level expression of most of the polyepitopes.

Answer and modification: We agree that this sentence was confusing and rephrased this for more clarity (lines 672-674)

Line 664-672: I would recommend removing or significantly revising this paragraph as it is highly speculative. If strong responses were induced by vaccination, they should still be observable above a background.

Answer and modification: We agree that this paragraph is highly speculative, but is nonetheless describing a relevant and well-documented feature of PRRSV that is likely to have had an effect of our read-outs. Because of this, we prefer to keep it, but have rephrased the section in lines 695-703

Round 2

Reviewer 2 Report

The authors have overall adequately addressed my concerns and comments.

Author Response

Thank you for your review :)
